# Transcriptomic Evidence Reveals the Dysfunctional Mechanism of Synaptic Plasticity Control in ASD

**DOI:** 10.3390/genes16010011

**Published:** 2024-12-25

**Authors:** Chao Kong, Zhitong Bing, Lei Yang, Zigang Huang, Wenxu Wang, Celso Grebogi

**Affiliations:** 1School of Systems Science, Beijing Normal University, Beijing 100875, China; kongch20@mail.bnu.edu.cn; 2Institute of Modern Physics, Chinese Academy of Sciences, Lanzhou 730000, China; 3School of Life Science and Technology, Xi’an Jiaotong University, Xi’an 710049, China; 4Institute for Complex Systems and Mathematical Biology, King’s College, University of Aberdeen, Old Aberdeen AB24 3UE, UK

**Keywords:** ASD, transcriptome, single cell sequence, signal transduction network, synapse plasticity, logic model, causal network

## Abstract

Background/Objectives: A prominent endophenotype in Autism Spectrum Disorder (ASD) is the synaptic plasticity dysfunction, yet the molecular mechanism remains elusive. As a prototype, we investigate the postsynaptic signal transduction network in glutamatergic neurons and integrate single-cell nucleus transcriptomics data from the Prefrontal Cortex (PFC) to unveil the malfunction of translation control. Methods: We devise an innovative and highly dependable pipeline to transform our acquired signal transduction network into an mRNA Signaling-Regulatory Network (mSiReN) and analyze it at the RNA level. We employ Cell-Specific Network Inference via Integer Value Programming and Causal Reasoning (CS-NIVaCaR) to identify core modules and Cell-Specific Probabilistic Contextualization for mRNA Regulatory Networks (CS-ProComReN) to quantitatively reveal activated sub-pathways involving MAPK1, MKNK1, RPS6KA5, and MTOR across different cell types in ASD. Results: The results indicate that specific pivotal molecules, such as EIF4EBP1 and EIF4E, lacking Differential Expression (DE) characteristics and responsible for protein translation with long-term potentiation (LTP) or long-term depression (LTD), are dysregulated. We further uncover distinct activation patterns causally linked to the EIF4EBP1-EIF4E module in excitatory and inhibitory neurons. Conclusions: Importantly, our work introduces a methodology for leveraging extensive transcriptomics data to parse the signal transduction network, transforming it into mSiReN, and mapping it back to the protein level. These algorithms can serve as potent tools in systems biology to analyze other omics and regulatory networks. Furthermore, the biomarkers within the activated sub-pathways, revealed by identifying convergent dysregulation, illuminate potential diagnostic and prognostic factors in ASD.

## 1. Introduction

Autism Spectrum Disorder (ASD) is a neurodevelopmental disorder [1,2,3,4] characterized by a range of clinical phenotypes and social features [5,6,7]. While a few subtypes of ASD can be determined by genetics in etiology [8,9,10,11,12], most are defined by social impairments and repetitive behaviors according to the fifth edition of the DSM manual [13,14,15]. Increasing evidence suggests that the genetic basis for ASD, a complex psychiatric disorder, is influenced by complicated interactions of multiple genes [16,17,18]. Scientific inquiries have further unveiled shared phenotypic traits in ASD, including neurophysiological abnormalities within neural circuits [19,20], synaptic density and pruning [21], and neuronal characteristics [22]. Notably, one distinctive endophenotype of ASD pertains to synaptic functional dysregulation [23,24,25,26,27] within the PFC, particularly involving changes in the plasticity of glutamatergic synapses [28,29,30,31], crucially linked to long-term potentiation (LTP) and long-term depression (LTD) [32,33,34,35,36]. Recent strides in next-generation sequencing within the realm of molecular biology [37], especially in omics technologies [38,39,40,41,42], have enabled large-scale investigations into the neurobiological dysregulation mechanisms of ASD at the genetic and molecular levels. In particular, transcriptomics [43,44,45,46,47,48] has emerged as a potent avenue for understanding the mechanistic underpinnings of ASD.

However, integrating omics data, especially across RNA and protein realms at the cellular level in ASD, remains challenging. Current studies primarily focus on genetic variations [2,49,50,51,52,53,54], isolated gene expression discrepancies [55,56], and genetic loci or gene associations [57,58]. To address this issue and gain a more comprehensive understanding of ASD’s molecular mechanisms, the concept of convergence related to a specific endophenotype has been introduced [59,60,61,62,63]. It poses a critical question: Do transcriptomics variations from genetic and early environmental risk factors ultimately converge at the protein level [64,65,66,67,68]? In response to this challenge, we develop a framework to transform the signal transduction network concerning synaptic plasticity-related phenotypes into the mRNA Signaling-Regulatory Networks (mSiReNs). It facilitates the analysis of signaling network dysregulation utilizing transcriptome-level data.

Our research utilizes single-cell nucleus RNA sequence data from PFC tissue. Then, we employ the Cell-Specific Network Inference via Integer Value Programming and Causal Reasoning (CS-NIVaCaR), derived from the CARNIVAL algorithm developed by Saez-Rodriguez et al. [69,70,71], to extract subnetworks specific to eight distinct cell types in ASD. Subsequently, we identified shared networks featuring EIF4E and EIF4EBP1 as key nodes responsible for translation control in the postsynaptic environment, referred to as core modules, within excitatory and inhibitory neurons. Further, we develop a computational approach called Cell-Specific Probabilistic Contextualization for mRNA Regulatory Networks (CS-ProComReNs) to identify dysregulatory sub-pathways in ASD. In doing so, we unveil four distinct patterns of sub-pathways within the core network from MAPK1 to EIF4E, encompassing MKNK1 (I), RPS6KA5 and EIF4EBP1 (II), EIF4EBP1 (III), and MTOR to EIF4E (IV), each with influence over synaptic translation in ASD. Notably, microRNAs, long non-coding RNAs (lncRNAs), and pseudogenes are found to regulate nodes within the core network.

Furthermore, we have confirmed the existence of regulatory relationships within mSiReN through protein interactions, as validated by the STRING database. This research methodology extends beyond conventional correlation-based studies. It integrates RNA quantification data and directed signed regulatory information from protein signaling networks to explore the consistency of molecular dysregulation between RNA and protein in the context of ASD’s translational control about synaptic plasticity. The amalgamation of mSiReN, NIVaCaR, and ProComReN offers a robust methodological framework, particularly applicable when handling extensive omics data for quantitative investigations into the causal molecular mechanisms underlying psychiatric disorders.

## 2. Materials and Methods

### 2.1. Signal Transduction Network About Translation Control of Postsynapse Plasticity

Each node and edge in our signal transduction pathway is described concisely and supported by corresponding references, provided in Appendix A for nodes and edges. Collectively, these regulations governing translation control in glutamatergic neurons, triggered by the activation of postsynaptic receptors, can be summarized into several pathways: PI(3)K-PDK-AKT, Ras-ERK(MAPK)-RSK/MSK, TSC1/2-RHEB-mTOR, and the FMRP-EIF4E-CYFIP1 complex, as depicted in the red dotted box in Figure in Section 3. Specifically, we focus on the translation control processes of proteins associated with synaptic plasticity, influenced by the activation of various neurotransmitters and synaptic stimuli at the postsynaptic receptors.

Regarding the upstream signaling nodes relevant to ASD, they are implicated in the dysregulation of glutamatergic synapses, particularly with GluR crosstalk, during the induction of postsynaptic plasticity. Glutamate activation of AMPA and NMDA receptors (NMDARs) plays a significant role in altering the cellular function of postsynaptic neurons and is crucial for synaptic plasticity. Metabotropic glutamate receptors (mGluRs) regulate mRNA translation, which is necessary for long-lasting forms of synaptic plasticity. NMDARs and metabotropic glutamate receptor 5 (mGluR5) employ distinct pathways for long-term depression (LTD) induction, but both converge on the internalization of AMPA receptors (AMPARs) [72], albeit potentially targeting different populations. The synaptic plasticity of neuronal cells involves the degradation of ARC by UBE3A, impacting the trafficking of AMPA receptors. NMDA receptor activation triggers calcium-dependent signaling, leading to the stimulation of CaMKII and modifications in AMPA receptor function and actin reorganization.

Upstream nodes activate downstream signaling nodes. Specifically, the activation of mGluRs initiates multiple signaling cascades, including the PI(3)K-AKT-mTOR pathway and the MAPK pathway [32,73,74], to regulate mRNA translation. Moreover, BDNF regulates mRNA translation by binding to the receptor TrkB, initiating signaling cascades such as the PI(3)K-AKT and Ras-MAPK pathways. In the PI(3)K-AKT pathway, PI(3)K acts as a lipid phosphatase and converts PtdIns(4,5)P2 to PtdIns(3,4,5)P3. PDK1, in conjunction with PtdIns(3,4,5)P3, activates AKT, subsequently inhibiting TSC1-TSC2. TSC1-TSC2 serves as a GTPase activating protein for the GTPase Rheb [32], promoting the conversion of active Rheb-GTP to the inactive GDP-bound form. Activation of Rheb-GTP ultimately leads to the induction of mRNA translation through MTOR.

Downstream signaling nodes continue to conduct and influence translational control in postsynaptic cells. PTEN and TSC1-TSC2 have been associated with ASD and mutations in Ras, Raf, and MEK1-MEK2 [32]. The balance between phosphatidylinositol-30-kinase (PI3K) and PTEN activities in the PI3K and PTEN pathway maintains basal levels of synaptic transmission and regulates the normal functioning of synapses. However, disrupted PTEN activity in individuals with ASD can lead to inadequate synaptic depression during plasticity events, contributing to the disorder’s pathogenesis. Furthermore, different molecular regulatory mechanisms of translation control are involved in early-stage and late-stage long-term potentiation (LTP). MNK regulates the CYFIP1/FMRP translation repressor complex in early-stage LTP and is involved in the regulation of 4E-BP2 and dendritic protein synthesis in late-stage LTP [5,75].

### 2.2. The Pipeline of Constructing the mSiReN

#### 2.2.1. Initial Nodes and Final Nodes from Empirical Knowledge

We examine the composition of every protein or complex in the collected signal network using the HGNC/UniProt database (https://www.uniprot.org/, accessed on 18 June 2022) based on “gene family” and “Alias Name”. In this process, we verify the nodes of precursor mRNA for these proteins (for more details see Appendix A). Subsequently, we construct the links between their precursor RNAs through the molecular interaction database using this information. Specifically, we first identify the initial nodes (BDNF, IGF1, RAS (HRAS/NRAS/KRAS), PI3K, and CAMK2) and the final node (EIF4) of the postsynaptic signaling transduction network. We extract the precursor mRNA of these nodes based on empirical knowledge and literature research. We determine the following source and target nodes’ corresponding labels and gene names in the signal network (see Appendix A).

#### 2.2.2. Match with Annotation Database

Protein complexes can be excluded from the analysis, such as TSC1/2, which is not present in the matching network. Subunits of PI3K, such as PIK3CA/PIK3CG, can be used instead. Then, we match the information of these nodes with the signal annotation databases, SIGNOR and SignaLink (see Appendix A), and other node information to help determine which pathways the proteins of interest correspond to. Specifically, we select the SignaLink pathway and SIGNOR databases, filtering the former for the “Receptor tyrosine kinase” pathway. For the latter, we filter for “AMPK Signaling”, “Glutamatergic synapse”, “MTOR Signaling”, and “PI3K/AKT Signaling”.

#### 2.2.3. Match with Interaction Database

Finally, we match the molecular interaction databases after determining the initial and final nodes.

Selection of Signal Annotation Databases and Key Signaling Pathways: Initially, we match the nodes from the literature-mined network with annotation databases for signal pathway-related information.Matching the initial input and terminal nodes with other network nodes in the annotation databases to find molecular interaction relationships. Only nodes matched with signal annotation databases from the literature-mined network are included in the final network. Matching involves identifying upstream signaling and neurotransmitter receptor nodes in the source nodes of the database, as well as translation control nodes in the target nodes. Other signal nodes are also matched.Choosing the direct interactions of signal nodes: Searching for regulatory subnetworks in the signal-regulated network containing the nodes of interest.

### 2.3. The High-Throughput scRNA-Seq Data and the Preprocessing of DEGs

#### 2.3.1. Single-Cell RNA Sequencing Data

We use 41 brain tissue samples from the PFC, including 16 control subjects and 15 ASD patients [56]. Single-cell RNA sequencing (scRNA-seq) data processing uses the 10X Genomics CellRanger software v7.0.0 [56]. The data are filtered based on individual expression matrices containing the number of Unique Molecular Identifiers (UMIs) per nucleus per gene. Nuclei are filtered to retain those with at least 500 genes expressed and less than 5% of total UMIs originating from mitochondrial and ribosomal RNAs. The individual matrices are combined, and UMIs are normalized to the total UMIs per nucleus and log-transformed.

In our study, we analyze 62,166 cells from the PFC region. Among these, 32,019 cells are from individuals with ASD, and 30,147 are from control subjects. Specifically, 35,356 cells are from BA9, 3849 cells are from BA46, and the remaining 22,961 cells are from the PFC region but are not explicitly categorized into BA9 or BA46. These cells are labeled with “_PFC” (including 10,472 ASD cells and 12,489 control cells) or “_PFC_Nova” (all control cells). In summary, if we consider the PFC region as a whole instead of only focusing on BA9 (BA9: 35,356 cells, ASD: 21,547, and Control: 13,809), we have an additional 26,810 cells. However, most of these cells are control cells from the PFC region (16,338 cells), and there are no ASD cells in the BA46 region. The detailed information for cell numbers in ASD and CTL can be found in Appendix A. The gene types utilized in our research, comprising 19,225 protein-coding genes, are sourced from the Human Gene Nomenclature Committee (HGNC) at www.genenames.org, which assigns unique symbols and names for various human loci, including protein-coding genes, non-coding RNA genes, and pseudogenes.

#### 2.3.2. Differential Expression Analysis

We use the R-language “Limma” package to analyze the differentially expressed mRNA profile data in the PFC, with the threshold of $\log_{2}\mathrm{FC}$ absolute value for filtering the eight-cell types set as 0.3, while ensuring that their *p*-value (*t*-statistic [76], see Appendix A for details) is less than 0.05. Fold Change (FC) characterizes the relative expression level of samples of interest to that of the control samples. RNAs with FC are more significant than 0.3 and correspond to an upregulated gene, while those less than minus 0.3 are associated with a down-regulated gene. The volcano map in Appendix A shows the individual RNA expression.

We conduct a comprehensive analysis of a total number of 684 genes for the RNA expression data. To identify DE genes, we apply the following thresholds: a *p*-value less than 0.05 and an absolute log FC more significant than 0.3. These thresholds are chosen to ensure statistical significance and a substantial magnitude of differential expression. The numbers of DEGs in different cell types are listed in Appendix A.

### 2.4. CS-NIVaCaR: Cell-Specific Network Inference via Integer Value Programming and Causal Reasoning

Gene expression profiling provides valuable insights into cellular processes, but uncovering the underlying regulatory processes that drive protein expression changes remains a maze. NIVaCaR addresses this challenge by deriving network architectures from high-throughout sequence gene expression data, specifically the DEGs from contrast experiments.

A key feature of NIVaCaR is its utilization of a Prior Knowledge Network (PKN) to represent protein connectivity knowledge. Unlike traditional approaches that rely on undirected and unsigned protein–protein interactions (PPIs), NIVaCaR leverages directed and signed signaling reactions, enhancing the interpretability and predictive power of the results. The proposed NIVaCaR ILP formulation is based on the formulation by Melas et al. [69], modified at critical points to address the computational complexity of single-cell signaling networks. It attempts to combine ASD/CTL gene expression data upon perturbation with the interrogated upstream nodes (or transmitter receptor) and identify the module that appears to be an anomaly. Of all the subsets of the PKN that achieve the desired targets for gene connectivity, the ILP algorithm selects the one numbering the fewest nodes.

One notable improvement in NIVaCaR is an inference for network activity overall rather than focusing on the upstream node from expression data of downstream nodes in the network. Unlike the original method, CARNIVAL focuses on gene expression regulation for signaling networks from diverse sources, including transcription factor targets and pathway signatures from other datasets. Our method is designed for the signal transduction-like mRNA Signaling-Regulatory Network (mSiReN) from single-cell RNA sequence data to reveal the activated pathway of translation control part across cell types.

In our work, a signed and directed postsynapse mSiReN, retrieved from a signal transduction network, affects the protein/complexes production of AMPA-related LTP or LTD. This PKN network has 99 signed and directed edges, connecting 44 nodes from multiple curated resources: Omnipath, Signor, Reactome, and Wikipathways. With its enhanced network contextualization capabilities and utilization of a directed and signed PKN, NIVaCaR opens new avenues for causal reasoning and network analysis in biological research. The identified pathways are functional subsets of all mSiReN and originate at the transmitter receptor, span across the signaling regulations, and go through the affected translational control.

#### The Objective Function for NIVaCaR

We implement the causal reasoning Integer Linear Program (ILP), formulated by the objective function in Equation (Equation 1) together with the constraints. The formulation aims to identify the minimum subset of *G* that minimizes the mismatch between measurements in a specific cell type and model predictions. Thus, the objective function Equation (Equation 1) is defined as:(1)$$\min\sum \alpha |x_{j,k}-c_{j,k}|+\sum \beta x_{j,k}^{+}+\sum \beta x_{j,k}^{-},$$
where the parameter α refers to the mismatch penalty, and β to the node penalty. The multiple α-to-β ratios are recommended for a value between 0.03 and 0.5 [70]. The objective function prioritizes the network in which the node activities $x_{j,k}$ explain the corresponding observed discretized measurements from cell types $c_{j,k}$. In contrast, the overall number of nodes in the network is minimized through the sum of activities ($x_{j,k}^{+}$ and $x_{j,k}^{-}$) for each node j for measurement k (cell type) in the network.

This approach, combined with the efficient handling of this information by the Integer Linear Programming (ILP) algorithm, sets NIVaCaR apart for network inference. Building upon the Causal Reasoning method introduced by Melas et al. [69], NIVaCaR offers enhanced capabilities for causal network contextualization. NIVaCaR needs a PKN and differential gene expression data. The PKN comprises causal protein interactions, while the gene expression data can be derived from microarray or RNA-seq experiments. These inputs are discretized to generate ILP constraints, and the actual continuous values are used to weigh and select causal links in the network (for additional information about NIVaCaR, see Appendix A).

### 2.5. CS-ProComReN: Cell-Specific Probabilistic Contextualization for mRNA Signaling Regulatory Networks

The Procedure of the ProComReN Algorithm

Prior knowledge network and experimental data from PFC tissue are combined to generate a network optimization problem. After the optimization process, the properties of the optimal network are then analyzed. Logic networks are optimized with semi-quantitative states between 0 and 1 at quasi-steady state. A probabilistic logic network approach represents the state of a node in a semi-quantitative range between 0 and 1 while it contains only one probability parameter per interaction.

In modeling logical networks, ProComReN represents biological regulatory systems as a dynamic Bayesian network (DBN), which is a directed graphical model defined by the set of n nodes with $X={\left[0,1\right]}^{n}$ and the probability distribution $P\left(X_{t}|X_{t-1}\right)=\Pi_{i=1}^{n}P\left(X_{t}^{\left(i\right)}|Pa\left(X_{t}^{\left(i\right)}\right)\right)$, where $X_{t}^{\left(i\right)}$ denotes the *i*’th node at time *t* and $Pa(X_{t}^{\left(i\right)})$ represents the parents of $X_{t}^{\left(i\right)}$. The structure of the network implicitly formulates these conditional probabilities. The different nodes represent the different molecules of the system, with a value corresponding to the degree to which these molecules exist in their active form (for example, phosphorylated proteins). These node values can be understood as the proportion of the molecules in the system being active or the probability of a randomly chosen molecule being active at time *t*.

In the ProComReN framework, each molecular interaction is formulated as a logical predicate associated with a weight quantifying the relative importance of that specific interaction. We model different biochemical interactions with two types of edges: positive and negative edges connect activators and inhibitors to their downstream targets. Each edge is associated with a weight $k_{j}^{\left(i\right)}$ representing the relative influence of the upstream node to the downstream node. Because our modeling framework is grounded in Bayesian theory, the weights must obey the law of total probability. For each node $X^{\left(i\right)}$ having a set $j_{+}$ of *m* activating functions, we ensure the sum of activating weights $\sum_{j_{+}=1}^{m}k_{j_{+}}^{\left(i\right)}=1$. Similarly, as weights of inhibiting interactions materialize the relative inhibition of upstream nodes, for nodes having a set $j_{-}$ of *l* inhibiting functions, we ensure that $0⩽\sum_{j_{-}=1}^{l}k_{j_{-}}^{\left(i\right)}⩽1$.

The Steps of the ProComReN Algorithm

I.Model Initialization:Providing the precursor RNA network as PKN.The combination of the activity/inhibitor of input nodes as the experimental conditions.The RNA expression of molecules as node measurements.Initialize the normally distributed values of the nodes, except for the input nodes.Assign random initial weights to the edges.II.Computation of Steady-State:Update the values of the nodes iteratively according to the DBN formulation:
(2)$$X_{t}^{\left(i\right)}=\sum_{j_{+}=1}^{m}k_{j_{+}}^{\left(i\right)}Pa{\left(X^{\left(i\right)}\right)}_{t-1}^{\left(j_{+}\right)}*\left(1-\sum_{j_{-}=1}^{i}k_{j_{-}}^{\left(i\right)}Pa{\left(X^{\left(i\right)}\right)}_{t-1}^{\left(j_{-}\right)}\right).$$Compute the expected value of each node’s probability distribution based on the values of its parent nodes and the associated weights.III.Contextualization with Experimental Data:Compare the Mean Squared Error (MSE) between the estimated and normalized measured values.Define an objective function: $\mathrm{Min}\left(\sum_{n=1}^{N}{\left(x_{n}-{\hat{x}}_{n}\right)}^{2}\right)$Use a gradient-descent algorithm (e.g., fmincon, function minimization with constraints, with the interior-point method) to optimize by adjusting the weights.Iterate the optimization process until convergence or a stopping criterion is met.

### 2.6. The Strength of the Sub-Pathway (SSP) and Abnormality Index of the Sub-Pathway (AISP)

The weight of the edge in mSiReN from ProComReN represents the relative influence of the upstream node to the downstream node. For a node $X^{\left(i\right)}$ that has a set of *m* activating edges, denoted as *j*, and each edge assigned a weight, denoted as $k_{j}^{\left(i\right)}$. The sum of the activating weights $\sum_{j=1}^{m}k_{j}^{\left(i\right)}$ must equal 1. It means the weights associated with activating interactions should collectively account for the total influence on the downstream node. For a specific sub-pathway, we define the Strength of the Sub-Pathway (SSP) as:(3)$$SSP\;=\;\prod_{i=1}^{n}\left\{\sum_{j=1}^{m}k_{j}^{\left(i\right)}\right\}.$$
The value *n* is the total number of all nodes on the interested sub-pathway from input to the endpoint. The Abnormality Index of the Sub-Pathway (AISP)

For ASD is:(4)$$AISP\;=\;\prod_{i=1}^{n}\left\{\sum_{j=1}^{m}k_{j}^{\left(i\right)}*{\left|log_{2}\left(FC\right)\right|}_{j}^{\left(i\right)}\right\},$$
where the FC value is a fold change of DEGs from the comparison between ASD and CTL. AISP score represents the total impact upon one node from all upstream signals in a specific path. It exhibits the pattern diversity of sub-pathway dysregulation through signal transduction networks across various cell types in ASD.

### 2.7. The Framework of Analysis

Our articulated framework for obtaining the convergent evidence for the translation-related endophenotype of synapse plasticity analysis combines the following methods:

We identify the related signal transduction network, and construct the mSiReN network, using NIVaCaR to extract the core module/network, and training the quantitative model ProComReN to discover ASD activated patterns across cell types. A flow chart of these methods is illustrated in Figure 1. Each method is described in Section 2.

The overall framework for transforming the signaling transduction network into mSiReN is depicted in Appendix A. The construction process of mSiReN using databases is shown in Appendix A. A comparison of the ProComReN method in computational biology is illustrated in Appendix A.

### 2.8. Computational Packages and Database

We carry out data processing and statistical analysis using the R language (version 4.2.0), execute the NIVaCaR algorithm through “CARNIVAL”, extract interaction information from “OmnipathR”, carry out the ProComReN model with the help of “CellNOptR” and “CNORprob”, analyze differentially expressed RNAs by using the “Limma” package, and plot the heatmap of the enrichment analysis using the “heatmap”. The various signal and RNA regulation networks are visualized via Cytoscape (v3.10.0). Finally, we perform the gene ontology and KEGG functional enrichment analysis using the online tool Metascape (http://metascape.org, accessed on 2 July 2023).

## 3. Results

### 3.1. Signal Transduction Network and mSiReN

#### 3.1.1. Signal Network of Glutamate Synaptic Plasticity

ASD encompasses various phenotypes, and the dysregulation of synaptic plasticity and synaptic function is strongly associated with ASD risk genes. This endophenotype serves as a foundational point for modeling the molecular dysregulation mechanisms of ASD. In addition, Differential Expression Genes (DEGs) of ASD from scRNA-seq data are enriched in chemical synapses and postsynaptic regulation. For detailed information, please refer to Appendix A. Abnormalities in synaptic plasticity are significant neuropathological features in ASD, which further influence synaptic pruning and excitatory/inhibitory equilibrium.

To investigate the molecular mechanisms underlying synaptic dysfunction, focusing on glutamate neurons in ASD, we manually assemble a postsynaptic signal regulation network (Figure 2A). Detailed information for each node can be found in Appendix A, and information for the edges is provided in Appendix A. This network primarily encompasses well-known ASD-related signaling pathways, including RAS-MAPK-TSC, PI3K-AKT-mTOR, and translation control involving 4EBP and EIF4E. The network is categorized into three parts: upstream neurotransmitter and receptor components, signal regulation units, and downstream translation control (Figure 2A). Additionally, some regulons without input edges are labeled in red in Figure 2A. The network illustrates how NMDAR, mGluR1/5, and growth factors influence protein translation through the signal regulation section, ultimately leading to the modulation of long-term potentiation (LTP) or long-term depression (LTD) by AMPA at glutamatergic postsynapses.

#### 3.1.2. The Construction Pipeline of mSiReN

We transform the Protein–Protein Interaction (PPI) signal network into its mRNA Signaling-Regulatory Network (mSiReN), corresponding to the original signal transduction network. Refer to Appendix A for details about the construction process. The nodes in mSiReN exhibit a strong relationship with synapse structure and function, as determined by Synaptic Gene Ontologies (SYNGO, https://www.syngoportal.org, accessed on 2 July 2023) (see Appendix A). This transformation allows us to assess imbalances within the network using extensive RNA sequence data. To construct this RNA network (see Figure 2B), we rely on significant databases in three steps: (1) Protein and Complex Databases: We utilize HGNC and UniProt, which provide information about the composition of each protein node. (2) Annotation Database: SignaLink pathway and SIGNOR (refer to Appendix A) help us to select ASD-related consensus pathways, such as “Receptor tyrosine kinase” and “WNT” in the SignaLink pathway, as well as “AMPK Signaling”, “Glutamatergic synapse”, “MTOR Signaling” and “PI3K/AKT Signaling”. (3) Interaction Databases: We utilize Omnipath and NPinter to establish connections with regulatory directions in the protein signal network. All interaction details of the mSiReN structure are presented in Appendix A.

### 3.2. The Cell-Type-Specific Activated Sub-Networks and NIVaCaR

To investigate the activation of subnetworks in individuals (see Figure 2C), we employ a logical causal reasoning model known as Cell-Specific Network Inference via Integer Value Programming and Causal Reasoning (NIVaCaR) (see Appendix A) across neuronal types. Appendix A presents all activated nodes and edges for all eight cell types. This algorithm aids in identifying plausible pathways within the signal network by considering the directionality of interactions and utilizing binarized Fold Change (FC), which results from comparing the gene expression of ConTroL (CTL) and ASD samples (Appendix A lists all the results of the DE analysis).

By extracting the shared network from excitatory neurons (see Figure 3A) and inhibitory neurons (see Figure 3B), we identify the EIF4EBP1 and EIF4E regulation pairs as a core module within the translation modules, playing a pivotal role in each subnetwork discovered. For example, HRAS and NRAS in the upstream signals promote PIK3CA in L23 (see Appendix A), PIK3CG in L4, and PIK3CG in L56/L56CC. In the downstream part, TSC2 stimulates mTOR, activating EIF4EBP1 in L23. In L4, L56, and L56CC, AKT is identified as the stimulus for EIF4EBP1.

In inhibitory neurons (see Appendix A), we recognize similar activation patterns in INPV compared to L4, with NRAS and PIK3CA exerting influences, followed by AKT1 affecting EIF4EBP1 and EIF4E. In INVIP, apart from AKT1, MTOR is identified as a regulator of EIF4EBP1, indicating a competitive relationship between these two factors. However, the core module has not yet revealed significant (Wilcoxon test) differential expression characteristics between CTL and ASD (see Figure 3C) for EIF4EBP1, except in L56CC cells, and (see Figure 3D) for EIF4E, without the L4 type.

### 3.3. The Activated Sub-Pathways and ProComReN

#### 3.3.1. The ProComReN Results

To further investigate the dysfunction within this signaling network (Figure 2A), we manipulated the RNA expression data of ASD to establish a dynamic quantitative model to uncover the etiology of ASD. We posit that the fuzzy logic model should computationally describe the transformation from CTL to ASD at the RNA expression data level within the signal network. Specifically, we treat CTL and ASD as two distinct conditions, designating their gene expression as two-time points. Subsequently, we construct the signal model ProComReN (see Appendix A for detailed information) to simulate the changes in nodes within mSiReN after combining several input nodes.

Considering gene expression and network topology (see Appendix A), we select SKT11, NF1, and KRAS as upstream and regulon components to serve as input nodes. We then trained all other nodes to fit the gene expression values at the two-time points. We also utilized perturbation experiment data from single-cell RNA sequencing to optimize the network’s parameters (see Appendix A).

For example, in the case of the L23 cell type, we illustrated the optimization model in Figure 4A with red edges, depicting the transition from CTL to ASD through variations in these nodes. (See Appendix A for a list of the best models across eight cell types.) Figure 4B demonstrates the stability of discrepancies between nodes in RNA expression data and the simulated values across different best training models of ProComReN. Notably, all nodes transition from a CTL state to an ASD state with a continuous value, which can be considered an activity or quantity influenced by stimulation or depression from upstream signaling nodes (Figure 4C).

#### 3.3.2. The Activated Sub-Pathways

NIVaCaR’s analyses have identified the EIF4EBP1 and EIF4E regulatory pair as the core module. Standard variation analyses of node expression for all cell types show that variation within excitatory or inhibitory cells is minor compared to between them. For more details, refer to Appendix A. Consequently, we segregate cell types into two main categories: excitatory (L, L23, L4, L56, and L56CC) and inhibitory (IN, INPV, INSST, INSV2C, and INVIP). To discern which upstream sub-pathways dysregulate this critical translation control node, we construct the core network by extracting the first neighborhood of the core module, which contains MAPK1, MKNK1, RPS6KA5, and MTOR. There are four patterns of sub-pathways: MAPK1-MKNK1-EIF4E (I), MAPK1-RPS6KA5-EIF4EBP1-EIF4E (II), MAPK1-EIF4EBP1-EIF4E (III), and MTOR-EIF4EBP1-EIF4E (IV).

We continued to analyze the activated pathway in the core network in excitatory/inhibitory neurons. Based on the results of our best quantitative models from ProComReN, a unique pattern exists for every single cell type. For the primary cell type L, as shown in Figure 5A, the variation of edges MKNK1 (17.8) and EIF4EBP1 (18.4) to EIF4E and RPS6KA5 (12.1) inhibiting EIF4EBP1 are notably high. Specifically, in L23, MAPK1 facilitates RPS6KA5, which subsequently inhibits EIF4EBP1, releasing the inhibition of EIF4EBP1 on EIF4E (II), ultimately promoting the expression of EIF4E. In contrast, in L4, MAPK1 directly facilitates EIF4E through MKNK1 (I). Similarly, MTOR accelerates EIF4E expression by double restraining through EIF4EBP1 (IV). Interestingly, L56 and L56CC exhibit almost consistent activated sub-pathways, promoting EIF4E via MKNK1 (I). However, the routes through EIF4EBP1 (II, III, and IV) are silenced, even though they have different upstream repressor nodes from MAPK1, RPS6KA5, or MTOR, respectively.

For the IN primary cell type, the edges exerting influence on EIF4EBP1 from MAPK1 (13.5), RPS6KA5 (19.3), and MTOR (13.4) are more unstable. Specifically, in INPV, EIF4E cannot be regulated by MAPK1 through any intermediate nodes (I, II, and III) but only through the MTOR-EIF4EBP1 (IV) sub-pathway. INPV shares the same activated sub-pathway with L4 but lacks the MAPK1-MKNK1-EIF4E (I) route. EIF4E in INSST can be activated through three pathways (I, II, and IV), but the effect from any one of them is minimal. INSV2C, like L56/L56CC, regulates EIF4E only through the MAPK1 and MKNK1 (I) sub-pathway rather than the pathways containing EIF4EBP1. Finally, INVIP exhibits a unique pathway through EIF4EBP1 directly via MAPK1 (III) in all eight cell types.

Except for INPV, all three other cell types possess the MAPK1-MKNK1-EIF4E (I) pathway, and, aside from INSV2C, the other three cells exhibit the phenomenon of inhibition attenuation on the EIF4EBP1 node through upstream signals from MTOR (IV), RPS6KA5 (II), or/and MAPK1 (III or/and IV).

#### 3.3.3. The Evaluation of Activated Sub-Pathways

In the core network, taking sub-pathway IV as an example, MTOR typically positively influences EIF4E through its interaction with EIF4EBP1. Specifically, the reduced inhibitory effect of MTOR on EIF4EBP1 leads to a diminished negative regulation of EIF4E. In the context of ASD, this disruption in translation control, which is associated with synapse plasticity, is a potential causative factor underlying ASD-related impairments. For instance, in L23 cells from individuals with ASD, the activated Strength of Sub-Pathway (SSP) (as defined in Equation (Equation 3) in the Section 2) for Sub-Pathway II, based on the best parameters in ProComReN, is significantly high, reaching approximately 32.7% (calculated as 0.73 (MAPK1-RPS6KA5) × 0.64 (RPS6KA5-EIF4EBP1) × 0.70 (EIF4EBP1-EIF4E) × 100%). All SSPs for the four sub-pathways (I to IV) in all eight cell types are listed in Table 1. Furthermore, we introduce an Abnormality Index of the Sub-Pathway (AISP), as defined in Equation (Equation 4), which represents the extent of dysfunction by combining it with the Fold Change (FC) value of RNA expression for all four sub-pathways affecting the core module in ASD (listed in Table 1). When considering the combined network topology drawn in Figure 5B,D from ProComReN, along with AISP or SSP values, we discern the activated sub-pathways for all eight neurons and their cell-specific dysfunctional patterns, as illustrated in Table 1.

### 3.4. Convergent Evidence on Translation Control of Synaptic Plasticity

#### 3.4.1. Convergence in Abnormal Non-Coding RNAs and Pseudogenes

A crucial aspect of RNA regulatory mechanisms involves the modulation of miRNA-recognition elements (MRE) [77,78] by other ncRNAs or pseudogenes [79,80,81,82], contributing to the trans-regulation of gene expression. Consequently, we investigated the interactions between mRNAs and lncRNAs, microRNAs, and pseudogenes using databases such as miRTarBase and Targetscan for microRNAs, LncBase for lncRNAs, and NPinter for all. This analysis reveals close associations between specific molecules and ASD, as illustrated in Figure 6A,C. Expression data for lncRNAs and pseudogenes are sourced from Velmeshev et al. [56], while microRNA expression data in the PFC region are obtained from Wu et al. [55]. Please refer to Appendix A for additional details on these ASD-associated molecules.

#### 3.4.2. The Reliability of Edges with Protein Interaction in mSiReN

The mSiReN is not limited to RNA regulation; each node in the network also corresponds to the protein network. To assess the correlation between RNA and protein, we input all the relevant RNAs from mSiReN into the STRING database to validate the reliability of our ProComReN model based on mSiReN (Figure 6B). Enrichment analysis on STRING, using the core network as an example (comprising EIF4E [P06730], EIF4EBP1 [Q13541], RPS6KA5 [O75582], MAPK1 [P28482], MKNK1 [Q9BUB5], and MTOR [P42345]), reveals that all seven edges in the core network have a Combined Score of over 90% (Table 2). Notably, interactions with lower scores, such as EIF4E-MAPK1, EIF4E-RPS6KA5, EIF4EBP1-MKNK1, MAPK1-MTOR, and MKNK1-MTOR, are not included. The only PPI interaction not found in mSiReN is MTOR-EIF4E, which the IV sub-pathway can potentially replace. MTOR indirectly promotes EIF4E by releasing the inhibition of EIF4EBP1. The corresponding reliability of each edge between RNAs in mSiReN and proteins in the signaling interaction is provided in Appendix A.

#### 3.4.3. Core Network with ADRI Score

Notably, within the core network, MTOR [83,84] (with a gene score of 2 and identified as a syndromic gene) and EIF4E [85] (with a score of 3) are considered autism susceptibility genes. Additionally, MAPK (MAPK3) and RPS6KA (RPS6KA2/3) share homologies, as indicated by the SFARI evaluation system (https://gene.sfari.org/, accessed on 2 July 2023). Nodes HRAS, NF1, PTEN, and TSC1/2 in mSiReN are also SFARI genes, each with a score of 1. Furthermore, RAC1 and RHEB are associated with various syndromes of ASD subtypes. (See Appendix A for more detailed information.)

## 4. Discussion

Integrating the diverse endophenotypes of ASD into a unified molecular explanation presents a considerable challenge. Despite significant progress in identifying ASD risk genes and genetic variations, elucidating the causal molecular mechanisms underpinning specific key endophenotype features remains paramount in the massive biological omics data era. Our research aims to clarify the fundamental causality behind ASD dysfunction related to synaptic plasticity and outline an overarching convergent mechanism from the RNA level to protein variation.

We gathered postsynaptic glutamatergic neuron-related signal networks to achieve this objective and used experimental data from PFC tissue. We employed an RNA interaction database to transform the protein signal network into mSiReN. Furthermore, we used the NIVaCaR method to calculate activated subnetworks within each cell type. We abstracted the core network formed by the first adjacency node of the core module EIF4EBP1-EIF4E, which regulates protein translation in the context of synaptic plasticity. Surprisingly, we uncovered the regulatory relationship between EIF4E and EIF4EBP1 in all excitatory and INPV and INVIP neurons. These two genes have wide-ranging associations with synaptic plasticity-related transducers, such as AMPA receptors within the translation control unit [75,86,87,88].

Subsequently, we reconstructed the experiment-like gene expression data of both CTL and ASD cases into dynamic changes at two different time points. We established a quantitative dynamic regulatory model called ProComReN to contextualize mSiReN for ASD cells. Refer to Appendix A for the relationship between the NIVaCaR and Pro-ComReN algorithms. This quantitative fuzzy logic model unveiled diverse sub-pathways of signal dysregulation, each with distinct patterns in excitatory and inhibitory neurons in the context of ASD. Finally, we assessed the reliability of this transformation from RNA to protein using STRING datasets, which demonstrated that all regulatory edges in the networks are highly dependable (see Figure 6B).

In summary, we have developed a network construction pipeline that transforms the protein signal network into its corresponding mRNA network. We leveraged gene expression data from single-cell RNA sequencing to train a quantitative causal model, shedding light on the dysfunctional sub-pathways in different ASD glutamatergic neurons. This framework of data processing and sequence databased numerical, boolean probabilistic network model provides a methodological reference for other omics-based mathematical models in computational biology and systems biology and offers a comprehensive exploration of the molecular causal mechanisms of ASD. This mechanism centers on RNA and protein molecular convergence related to translation-associated endophenotypes of synaptic plasticity rather than direct correlations, as understood in GWAS, WCGNA, phenotypic-gene associations, and other methods.

Notably, the genes associated with the signal transduction network for translation control in synaptic plasticity exhibit no clear connection (10 out of 406 DEGs in all cell types) with ASD risk genes identified in various sequence studies [41,89]. This finding suggests that ASD is not solely an aberration at the level of individual genes, such as base-pair variations, de novo mutations, or copy number variations (CNV). Regardless of the genetic variants implicated in ASD, the dysfunctional effects manifested at the RNA and protein levels [90]. Our analysis of the mechanisms at the transcriptomic and proteomic levels provided consistent evidence for ASD dysregulation, grounded in the principles of the central dogma. This perspective helped us to understand how genetic variations at the gene level translate into effects on neurocognition and behavioral patterns through gene expression, RNA regulation, and protein function.

ASD encompasses a multitude of genetic variations in terms of types and loci. Thus, modeling the molecular mechanisms of a specific phenotype in ASD necessitates the integration of multi-level causal mechanisms. Transcriptomics allows for the aggregation of changes in genetic risk and environmental factors related to ASD while modeling at the RNA level identifies potential dysregulations in protein function. A systems approach is essential for comprehensively studying ASD, a complex genetic disorder that cannot be fully understood through traditional bottom-up methods such as single-gene knockout experiments. Instead, systems biology offers a top-down approach focusing on global gene networks and system regulation principles. A quantitative causality model, with appropriate scale and elaborated regulation modeling at the RNA level, facilitates a comprehensive exploration of signal regulation dysfunction.

Compared to transcriptional control, which involves numerous transcription factors and exerts delayed effects on protein function, translational control offers a more suitable choice. This mode of control influences signaling pathways affecting synaptic function and has an immediate effect, contributing to rapid changes in synaptic responses. For our study, we concentrated on the glutamatergic postsynaptic regulation signals that directly affect translation control of plasticity, such as AMPAR and NMDA receptors, and various structural proteins, rather than focusing on transcription, which entails more intricate loops and dynamic factors.

It is worth noting that protein sequencing of brain tissue samples may not accurately reflect protein abundance in a living organism due to protein degradation and brain cells already dead, especially in human research. Currently, available proteomics and phosphorylation data related to signaling pathways linked to ASD synaptic plasticity dysregulation are limited. From an individual RNA-to-protein abundance perspective, the correspondence is uncertain, with an approximate probability of 30% if alternative splicing is not considered, or 70% [91,92,93] if the statistical correlation is considered. Therefore, utilizing RNA-level data represents a novel avenue for quantifying protein expression within signaling networks, underscoring the necessity of employing mSiReN and training transcriptome models (see Appendix A). Transforming the signal transduction network into mSiReN may pose potential instability. Nevertheless, it is a technically feasible method to construct a quantitative model by utilizing extensive transcriptomics data to describe the molecular mechanisms and provide an outline of convergent evidence related to gene-to-protein malfunction in ASD pathophysiology within the framework of large-scale omics research. Our methods take an exploratory step towards integrating causal mechanism modeling into molecular regulation, positioning us at the cutting edge of the field of computational systems biology.

## Figures and Tables

**Figure 1 genes-16-00011-f001:**
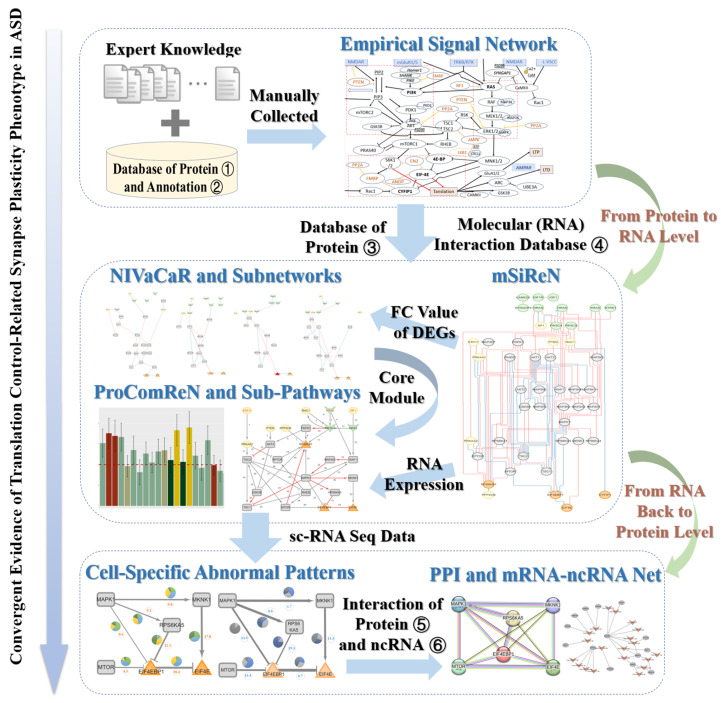
Framework for detecting ASD convergence in the translation-related endophenotype of synapse plasticity. Empirical signal network data for synapse plasticity control are manually collected and then transformed into mSiReN. Activated subnetworks in ASD are identified using extensive scRNA-seq data via NIVaCaR, and sub-pathways are discovered using ProComReN. The network structure of core pathway patterns is subsequently validated in PPI and targeted by other ncRNAs or pseudogenes. This research paradigm untangles information within the signal transduction network, decoding it within the RNA realm and then mapping it back to protein function. Databases utilized, as illustrated in the picture, include: (1) Protein Database: UniProt. (2) Annotation Database: SignaLink Pathway/SIGNOR. (3) Protein Database: HGNC/UniProt. (4) Molecular (RNA) Interaction Database: Omnipath. (5) Protein Interaction Database: String. (6) ncRNA Interaction Databases: NPinter, miRTarBase, Targetscan, and LncBase.

**Figure 2 genes-16-00011-f002:**
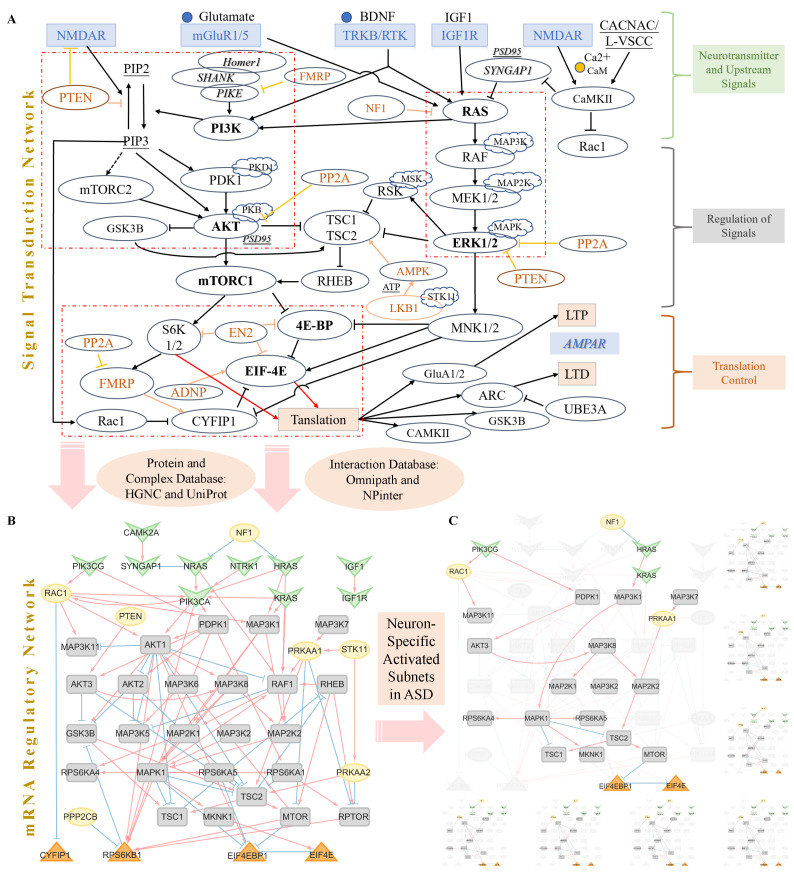
Signal transduction network, mSiReN, and cell-specific activated subnetworks in postsynaptic plasticity translation control. (**A**) This segment depicts a manually curated signal transduction network about the regulation of translation in postsynaptic plasticity within glutamate neurons. This network encompasses various components, including neurotransmitters and receptors such as NMDAR and mGluR1/5, as well as growth factors in the upstream region. Critical signal regulation units encompass the RAS-MAPK-TSC pathway and the PI3K-AKT-mTOR pathway. Notably, several regulons lacking input edges, highlighted in gold, contribute to the modulation of LTP or LTD mediated by AMPA receptors in the translation control section of glutamatergic postsynapses. (**B**) The mRNA Signaling-Regulatory Network (mSiReN) corresponds to the original signal transduction network (**A**). This network transforms proteins from the signal network into their corresponding precursor mRNAs, a process informed by databases such as HGNC and UniProt. This transformation elucidates the relationships established through interaction databases drawn from Omnipath and NPinter and information derived from the SignaLink pathway and SIGNOR annotation databases. (**C**) The activated subnetworks in ASD eight excitatory and inhibitory neuron types discovered by CS-NIVaCaR.

**Figure 3 genes-16-00011-f003:**
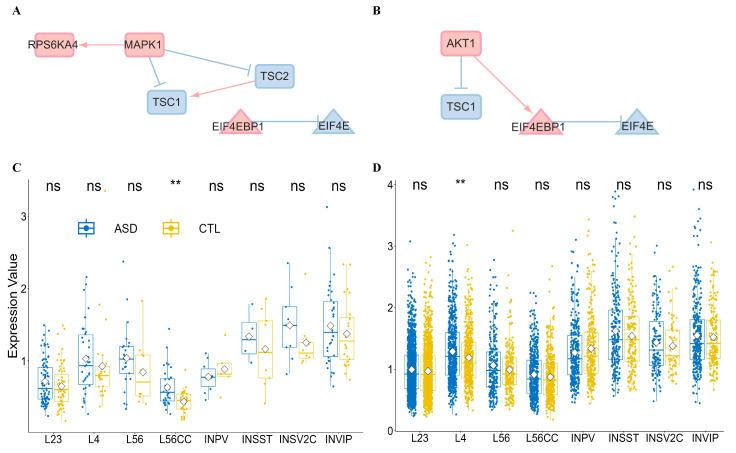
Core modules for excitatory and inhibitory neurons and expression characteristics in ASD. (**A**,**B**) Activated shared subnetworks are extracted from excitatory (**A**) and inhibitory (**B**) neurons by CS-NIVaCaR, revealing EIF4EBP1 and EIF4E regulation pairs as the core module within the translation control component. Activated nodes and edges are represented in red, while suppressed elements are depicted in blue. (**C**,**D**) Gene expression involving EIF4EBP1 (**C**) and EIF4E (**D**) is assessed in eight distinct cell types for CTL and ASD. The Wilcoxon test is employed for intergroup comparisons. Two asterisks “**” indicate the significance level less than 0.01. “ns” means no significant test result.

**Figure 4 genes-16-00011-f004:**
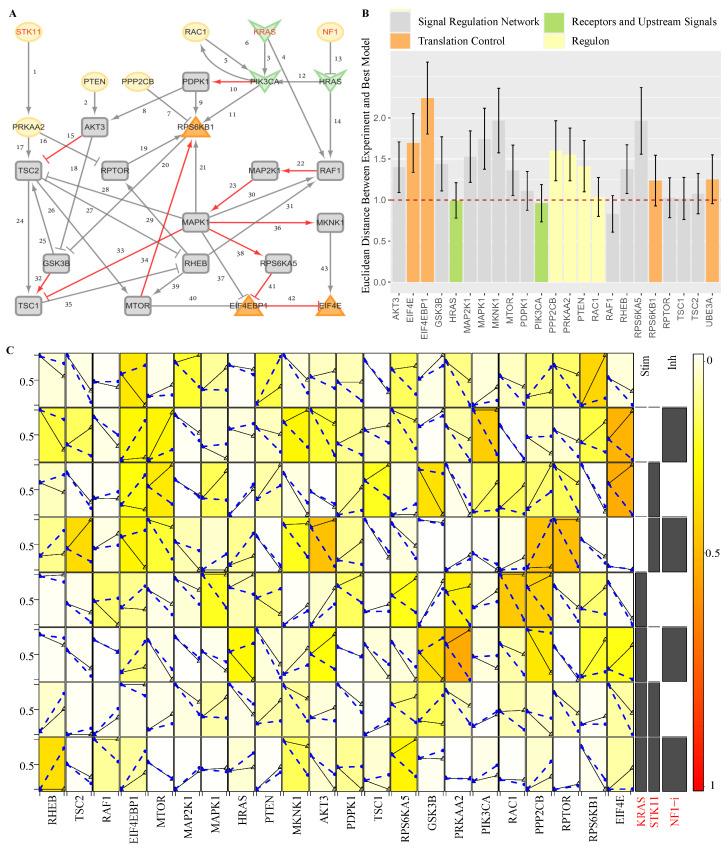
Establishment and optimization of the quantitative causal logic model ProComReN in L23 Cells of ASD. (**A**) The reduced network structure for the ProComReN algorithm. The activated pathway, regulated by inputs/stimuli from STK11, KRAS, and NF1 in L23, is depicted by red edges in the best results of ProComReN. (**B**) Accuracy statistics for ProComReN training models, compared to experiment-like data, across all four categorized nodes. (**C**) Changes in node values’ parameters, compared to experiment-like data at two-time points, under the combination of input signals within the best ProComReN model. The dashed blue line is the model simulation result, and the solid black line is the sample data.

**Figure 5 genes-16-00011-f005:**
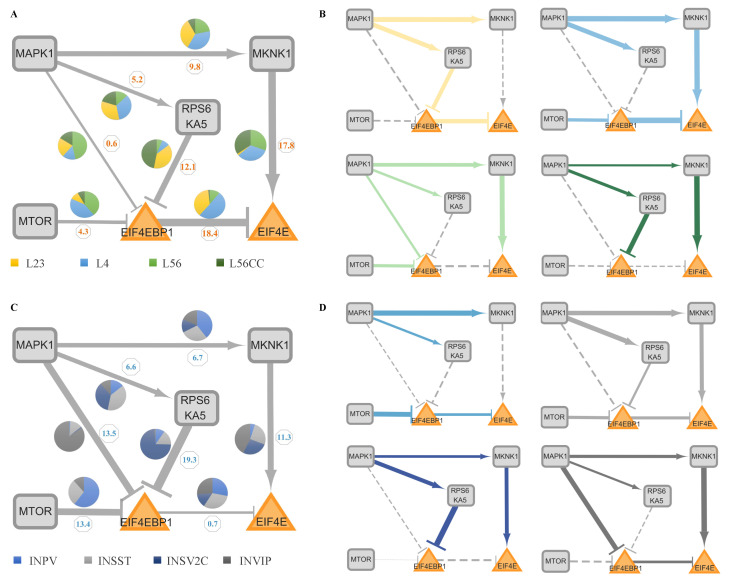
Core network and activated patterns of sub-pathways in eight cell types. (**A**,**C**) Variations in activated sub-pathways among four excitatory neurons (L23, L4, L56, and L56CC) (**A**) and four inhibitory neurons (INPV, INSST, INSV2C, and INVIP) (**C**) within the core network. Wider edges (indicated by numbers) correspond to more substantial changes within all excitatory (**A**) or inhibitory (**C**) cells, based on the ProComReN best model results. The pie chart illustrates the relative distribution of weights among different cell types for each edge. (**B**,**D**) Activated sub-pathway patterns in excitatory (**B**) and inhibitory (**D**) neuronal types in ASD. The width of the edges indicates the intensity of activation or suppression (weights exceeding 0.2) according to the best ProComReN model. Dashed lines denote inactivity.

**Figure 6 genes-16-00011-f006:**
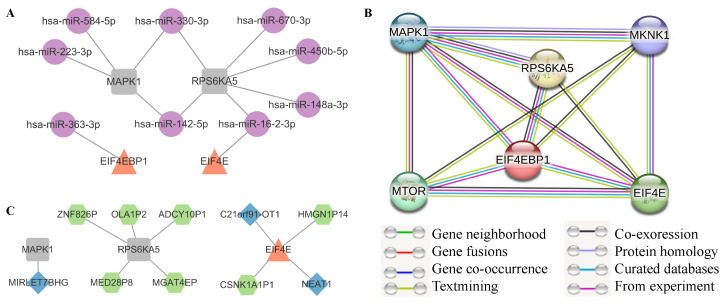
Convergent evidence of misaligned molecules in ASD. (**A**,**C**) Targeted microRNA (**A**) and lncRNA, along with pseudogenes (**C**), are identified within the core network using data from miRTarBase, Targetscan, LncBase, and NPinter databases. (**B**) The reliability of protein–protein interactions (PPI) within the relevant sub-pathways in the core network is assessed using data from the String database.

**Table 1 genes-16-00011-t001:** The index of SSP and AISP in the core network for eight excitatory and inhibitory neurons in ASD.

	I: MAPK1 (->) MKNK1 (->) EIF4E		II: MAPK1 (->) RPS6KA5 (-|) EIF4EBP1 (-|) EIF4E		III: MAPK1 (-|) EIF4EBP1 (-|) EIF4E		IV: MTOR (-|) EIF4EBP1 (-|) EIF4E		Sub-Pathway Pattern (ASD)
	SSP	AISP	SSP	AISP	SSP	AISP	SSP	AISP	
L23	4.1	31.4	32.7	33.69	9.8	12.7	7.7	11.89	II
L4	82.81	34.72	10.11	29.72	8.55	10.97	45.6	21.14	I and IV
L56	45.65	24.76	0.58	15.96	5.2	15.53	8.8	18.07	I
L56CC	19.74	25.25	1.01	23.45	0.3	15.51	0.27	15.39	I
INPV	9.9	21.47	1.12	13.02	0.33	9.29	27.72	20.55	IV
INSST	36.4	16.7	5.52	15.38	4.44	3.07	14.43	8.58	I, II and IV
INSV2C	18.56	11.71	10.66	15.34	0.17	3.92	0	3.76	I
INVIP	39.06	15.36	0.13	10.73	24.18	16.55	4.96	10.16	I and III

**Table 2 genes-16-00011-t002:** The correlation between core network and corresponding protein interactions from the STRING database. The edges of these blue marks are those that have a combined score higher than 0.9.

Node1	Node2	Homology	Coexpr	Experimentally Determined Interaction	Database Annotated	Automated Textmining	Combined Score
EIF4E	MAPK1	0	0.062	0.127	0	0.438	0.5
EIF4E	EIF4EBP1	0	0	0.996	0.9	0.994	0.999
EIF4E	MTOR	0	0.062	0.369	0.9	0.993	0.999
EIF4E	MKNK1	0	0	0.637	0.9	0.833	0.993
EIF4E	RPS6KA5	0	0.063	0	0	0.406	0.419
EIF4EBP1	MTOR	0	0	0.982	0.9	0.913	0.999
EIF4EBP1	RPS6KA5	0	0.049	0.213	0.9	0.578	0.964
EIF4EBP1	MAPK1	0	0	0.485	0.8	0.438	0.937
EIF4EBP1	MKNK1	0	0.055	0	0	0.556	0.563
MAPK1	MKNK1	0.582	0.056	0.721	0.9	0.787	0.98
MAPK1	RPS6KA5	0.642	0.062	0.319	0.9	0.388	0.939
MAPK1	MTOR	0	0.062	0.284	0	0.588	0.699
MKNK1	MTOR	0	0.062	0	0	0.391	0.404

## Data Availability

All relevant data are available from the authors upon request. All relevant computer codes are available from the authors upon request.

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
