# Peer review of "Transcriptomic Evidence Reveals the Dysfunctional Mechanism of Synaptic Plasticity Control in ASD"

_genes, 2024, doi:10.3390/genes16010011_

Round 1

Reviewer 1 Report

Comments and Suggestions for Authors

Journal: genes (ISSN 2073-4425)

Manuscript ID: genes-3362761

Type: Article

Title: Transcriptomic Evidence Reveals the Dysfunctional Mechanism of Synaptic Plasticity Control in ASD

The research title: “Transcriptomic Evidence Reveals the Dysfunctional Mechanism of Synaptic Plasticity Control in ASD” by Chao Kong et al, provides a very innovative and thorough examination of synaptic plasticity failure in ASD, utilizing powerful computational methods such as ProComReN and CS-NIVaCaR to identify cell-type-specific active sub-pathways. The combination of RNA-level regulatory networks with route analysis and metrics such as SSP and AISP offers a unique method for deciphering complicated molecular interactions. The evident distinction between excitatory and inhibitory neurons, as well as the extensive investigation of translation control mechanisms, provide important insights into ASD pathogenesis. Overall, the work is methodologically sound, scientifically relevant, and has the potential to shape future research and therapy approaches in neurodevelopmental diseases.

Minor comments:

1. Were the findings (e.g., deregulation of EIF4EBP1 and EIF4E) validated by experimental or clinical data?

2. Could a brief explanation of the innovative features of algorithms like as CS-NIVaCaR and CS-ProComReN be given to readers outside of computational biology? How do their principles and distinctiveness distinguish them from other approaches?

3. Line no.349-351: The summarized pathways (PI3K-PDK-AKT, Ras-ERK, etc.) are mentioned, Could you explain why these particular routes were chosen to be the study's focus? Are they particularly pertinent to the failure of synaptic plasticity in ASD?

4. Line no. 366-367: Although the calcium-dependent signaling cascade including AMPA receptor alterations and CaMKII has been identified, its precise relationship to the pathogenesis of ASD is not explained.
Could you elaborate on the connection between synaptic plasticity in ASD and the deregulation of this pathway?

5. Were independent datasets or experimental findings used to cross-validate the ProComReN algorithm's output? How well did they align, if at all?

6. In what way was the core module's "first neighborhood" defined? Did MAPK1, MKNK1, RPS6KA5, and MTOR get their selections from the dataset or from the literature alone?

7. Table 1: Why do L4 cells have the highest SSP value for Sub-Pathway I (82.81)? What does this indicate them about their function in synaptic plasticity in ASD.

Author Response

Response to Reviewer 1 Comments
We sincerely thank the reviewer for thoroughly examining our manuscript and providing very helpful comments to guide our revision.
Sincere thanks should be given to the reviewer for the constructive comments and suggestions. The responses to the comments are given below.

Comments and Responses
Comment 1: Were the findings (e.g., deregulation of EIF4EBP1 and EIF4E) validated by experimental or clinical data?
Response 1: This is an excellent question. Unfortunately, we have not conducted direct wet-lab experiments to validate these findings. We acknowledge that experimental validation is a critical next step to elevate our work to the highest standard. Our primary contributions lie in the theoretical and algorithmic innovations. However, we have made every effort to support our conclusions. Our study utilized transcriptomic data from post-mortem human brain samples, which limits our ability to perform in vivo validation. In this work, we identified significant interactions between EIF4EBP1 and EIF4E through PPI regulatory relationships. To corroborate these findings, we performed a comprehensive literature review and identified experimental studies that support our results, as cited in references [81–84]. These references are summarized in the manuscript's conclusion section:
1.    The fragile x syndrome protein represses activity-dependent translation through cyfip1, a new 4e-bp, Cell 134, 765–1042 (2008). PubMed
2.    Panja, D. et al., Two-stage translational control of dentate gyrus ltp consolidation is mediated by sustained bdnf-trkb signaling to mnk, Cell Rep 9, 1430–45 (2014). PubMed
3.    Aguilar-Valles, A. et al., Inhibition of group I metabotropic glutamate receptors reverses autistic-like phenotypes caused by deficiency of the translation repressor EIF4E binding protein 2, J Neurosci 35, 11125–32 (2015). PubMed
4.    Wiebe, S. et al., The EIF4E homolog 4EHP (EIF4E2) regulates hippocampal long-term depression and impacts social behavior, Mol Autism 11, 92 (2020). PubMed
________________________________________
Comment 2: Could a brief explanation of the innovative features of algorithms like CS-NIVaCaR and CS-ProComReN be given to readers outside of computational biology? How do their principles and distinctiveness distinguish them from other approaches?
Response 2: Thank you for this question, and we apologize if the complexity of the algorithms created obstacles to understanding our main contributions. The primary innovation of CS-NIVaCaR and CS-ProComReN lies in their applicability to single-cell datasets. As you may know, earlier methods for bulk data analysis were designed for single cell types, whereas single-cell datasets introduce greater complexity, making many traditional bulk methods inapplicable.
•    CS-NIVaCaR employs a simplified Boolean 0–1 logic algorithm to infer active regulatory interactions.
•    CS-ProComReN quantitatively analyzes inactive pathways within cells.
While these algorithms may not represent groundbreaking advancements in their computational underpinnings, they are tailored to the unique challenges of single-cell data. If readers are interested in exploring signal transduction pathways in the context of single-cell datasets, our methods offer a practical framework. Essentially, CS-NIVaCaR and CS-ProComReN were designed specifically to model signal transduction networks using single-cell transcriptomic data. They leverage Boolean logic and gradient optimization algorithms to identify optimal values, making them more suitable for single-cell data analysis than existing approaches.
________________________________________
Comment 3: Line 349–351: The summarized pathways (PI3K-PDK-AKT, Ras-ERK, etc.) are mentioned. Could you explain why these particular routes were chosen to be the study's focus? Are they particularly pertinent to the failure of synaptic plasticity in ASD?
Response 3: Thank you for pointing this out. Our selection of these pathways was based on extensive literature research and represents a key contribution of our work. We conducted a comprehensive review of ASD-related synaptic plasticity dysregulation and identified the main nodes and regulatory interactions, which are documented in Supplementary Files 1 and 2. These nodes and interactions were used to construct the ASD-specific post-synaptic regulatory network, as shown in Figure 1A. We agree that the current wording may be unclear, and we have revised Line 347 to include:
"Each node and edge in our signal transduction pathway is described concisely and supported by corresponding references, provided in Supplementary Files 1 and 2."
________________________________________
Comment 4: Line 366–367: Although the calcium-dependent signaling cascade including AMPA receptor alterations and CaMKII has been identified, its precise relationship to the pathogenesis of ASD is not explained.
Response 4: You are correct, and we acknowledge the limitation in our study's scope. Our analysis focuses on the molecular-to-molecular interactions within the context of ASD-related synaptic dysregulation. Determining the causal relationship between molecules and phenotypic outcomes would require extensive additional studies involving transcriptional, post-transcriptional, translational, and post-translational analyses across different cell types and tissues. This is beyond the scope of our current study.
To clarify, the relationship between AMPA receptor alterations, CaMKII, and synaptic plasticity in ASD is supported by classical literature, as detailed in Supplementary Files 1 and 2. Our work highlights statistical correlations rather than causal relationships between molecular features and ASD phenotypes.
________________________________________
Comment 5: Were independent datasets or experimental findings used to cross-validate the ProComReN algorithm's output? How well did they align, if at all?
Response 5: This is a fundamental question in computational biology. While our study focused on single-cell transcriptomic data analysis for ASD, we have yet to cross-validate the ProComReN algorithm due to the scarcity of post-mortem ASD brain samples. However, the algorithm was validated against the most suitable model parameters, as shown in Figure 3B.
________________________________________
Comment 6: In what way was the core module's "first neighborhood" defined? Did MAPK1, MKNK1, RPS6KA5, and MTOR get their selections from the dataset or from the literature alone?
Response 6: These nodes were identified from the network structure itself, as shown in Figure 3A. They were selected as part of the core module due to their first-neighbor connections with EIF4EBP1 and EIF4E, which were identified based on literature as ASD-related nodes.
________________________________________
Comment 7: Table 1: Why do L4 cells have the highest SSP value for Sub-Pathway I (82.81)? What does this indicate about their function in synaptic plasticity in ASD?
Response 7: SSP reflects the most pronounced pathways affecting EIF4E, as highlighted in Figure 4. Sub-Pathway I (MAPK1 → MKNK1 → EIF4E) showed higher SSP values in L4 neurons, indicating a potentially stronger association between synaptic dysregulation and this pathway in L4 neurons. However, these findings represent statistical correlations rather than causal relationships, as explained in Response 4.
Thank you sincerely for your thoughtful and detailed comments. Your feedback reflects the significant effort and expertise you have applied in reviewing our manuscript. It is evident that your broad knowledge across various specialized fields and your inclusive perspective on diverse research areas have greatly enriched the quality of our work. Your constructive suggestions have not only helped refine our study but also made our findings more accessible and comprehensible to readers. This collaborative process advances not just our individual research but also fosters inclusivity, openness, and, most importantly, rigor within the scientific community.

Reviewer 2 Report

Comments and Suggestions for Authors

This is a complex bioinformatics study of purported signaling abnormalities in excitatory and inhibitory neurons in the prefrontal cortex (PFC) samples from ~2 dozen subjects with either ante-mortem Autism Spectrum Disorder (ASD, a highly heterogenous sample diagnosed mainly on behavioral properties) and CTL. These samples are presumably age- and gender-matched. The authors carried out single nuclear RNA sequencing and analyzed gene expression data based on grouping of excitatory or inhibitory neurons using published algorithms. They grouped data using other algorithms (ie, CARNIVAL) that allowed them to examine expression levels of proteins in varying signaling modules to determine if expressions (mRNA levels) for regulatory translation proteins (ie EIF4E and its regulator EIF4EBP1) were altered significantly.

The bulk of this paper involves the “how” of characterizing the different signaling networks in subsets of excitatory or inhibitory neurons. From what I can gather, the only direct comparison between ASD and CTL samples is shown in Figure 2. There, EIF4E expression is significantly elevated in the ASD L56CC neuronal population (Fig. 2C) and EIF4EBP1 expression is significantly elevated in the ASD L4 cell population (Fig. 2D). These data do not appear to have been discussed in the text. Rather, there is extensive discussion about how these cell groups were defined (all in Supplemental Figures not available to this reviewer).

It is unclear what the thrust of this paper is. Is it about applying complex algorithms to gene expression data to sort out signaling systems in varying PFC neuronal populations, or is it a study of ASD, or possibly is it both? The authors have clearly done a tremendous amt of work (no description of the snRNASEQ approach, only the in silico work) and leave the reader wondering what their goals were.

I suggest that the authors move their algorithm work to Supplemental Methods and instead focus on differences between ASD and CTL in the text. The details presented in the algorithmic work would appeal mostly to experts in bioinformatics and not to non-expert readers who would be more likely interested in their findings in ASD. They need to describe their ASD and CTL PFC samples more extensively and then guide the reader through their complex analyses. At the moment, based on how they presented their findings, there appear to be two small changes in expression of mRNA’s for translation-regulating proteins in one of nine neuronal cell populations identified in the PFC’s of their samples.

Review of Supplemental Data in editor-provided PDF

1.     8 Supplemental notes

2.     22 Supplemental figures

3.     9 Supplemental Tables

The Supplemental Data is very extensive and more of a bioinformatics “book” about their approach rather than a traditional Supplement to a paper. My overall thoughts:

1.     I am not expert enough in bioinformatics to provide a comprehensive review of their approaches.

2.     The overall concepts behind their approaches are quite interesting. I support the qualitative concept of using scRNA-SEQ data to infer signaling mechanisms in differing populations of brain neurons. I predict that this approach will be used in many different diseases, since the technologies required are now available (both wet lab and in silico)

3.     They should either merge their Supplemental Data into a book or a very long article submitted to a specialty journal that would know of experts who could provide a meaningful review of their approaches.

4.     They should focus their IJMS_Gene paper on Autism Spectrum disorder (ASD) as analyzed using their approach/models (which are detailed separately). In order to do this, they would need to provide much more information about the collection of their postmortem human brain specimens. In particular, such collections must be IRB approved, show matching for postmortem intervals, gender, ages, etc.

5.     My original comment about the ambiguous “purpose” of this paper is confirmed by the length and complexity of their Supplemental data.

Author Response

Response to Reviewer 2 Comments
We sincerely thank you for your insightful comments on the manuscript. First, we acknowledge that this work spans a significant breadth, as it represents our investigation into a highly challenging research question. The study is particularly demanding because neurodevelopmental disorders like ASD (autism spectrum disorder) encompass a multitude of phenotypic characteristics, and establishing links between these phenotypes and their underlying genotypes is extraordinarily difficult. However, advancements in single-cell omics technologies have provided an unprecedented opportunity. Leveraging our expertise in mathematics, physics, and algorithm development, we aimed to delve into the molecular mechanisms of ASD.
Faced with this formidable challenge, we adopted a multi-faceted approach. This included an extensive literature review to identify pathways associated with synaptic plasticity impairments in ASD, selecting appropriate algorithms, and conducting rigorous computational simulations. Nevertheless, as you have noted, we lack direct experimental validation (wet-lab studies) to substantiate the gene expression differences between normal and ASD-affected groups. As a result, our focus has been heavily skewed towards algorithmic precision and methodological rigor, rather than providing a concise and profound interpretation of these findings.
Another important point relates to the theme of this work. As you observed, the manuscript may initially appear thematically ambiguous—whether it is primarily focused on molecular pathway analysis or algorithm development. We recognize the challenge this poses. The interdisciplinary nature of this study, which investigates molecular dysregulation in ASD, required us to develop new algorithms tailored for the rapidly growing field of single-cell omics. Simultaneously, understanding ASD's complex pathophysiology led us to conduct detailed analyses of pathways and molecular regulation. This inevitably resulted in a seemingly broad and intricate presentation.
We aimed to provide readers with a comprehensive demonstration of how cutting-edge omics technologies and algorithms can be integrated to explore disease mechanisms. Thus, we opted not to separate the algorithmic aspects from the molecular pathway analyses, despite the potential for confusion among readers from specialized fields. Alternatively, this work can be viewed as a holistic effort to explore synaptic plasticity impairments in ASD by combining molecular insights and computational innovations.
Furthermore, after considerable deliberation, we decided to emphasize the algorithmic contributions in the abstract while drawing conclusions about ASD-related molecular dysregulation. Readers primarily interested in the molecular biology details of ASD can refer to the supplementary materials, which provide additional in-depth information on the pathways and regulatory mechanisms explored in this study.

We sincerely hope that this revised manuscript has addressed all your comments and suggestions. We appreciated for reviewers' warm work earnestly, and hope that the correction will meet with approval. Once again, thank you very much for your comments and suggestions.
We would like to thank the referee again for taking the time to review our manuscript.